# OpenReview forum: "XR-1: Towards Versatile Vision-Language-Action Models via Learning Unified Vision-Motion Representations"
_ICML.cc/2026/Conference — ICML 2026 spotlight_

### Official Review · Reviewer_CvQt · 2026-03-12

**Soundness:** 4
**Presentation:** 4
**Significance:** 3
**Originality:** 3
**Overall Recommendation:** 5
**Confidence:** 3

**Summary:**

This paper presents XR-1, a VLA framework that tackles the precision gap between high-dimensional observations and low-level motor commands, as well as the domain gap across heterogeneous data sources, including diverse robot morphologies and actionless human videos. The main contribution is Unified Vision-Motion Codes (UVMC), a discrete latent representation learned through a dual-branch VQ-VAE with a shared codebook and a KL-based cross-modality alignment loss, which jointly encodes visual dynamics and robotic motion. XR-1 employs a three-stage training paradigm and demonstrates flexibility through both a full-scale variant built on $\pi_0$ and a lightweight variant built on SwitchVLA. The authors validate across six robot embodiments and over 120 manipulation tasks with more than 14,000 real-world rollouts, reporting consistent improvements over baselines, alongside generalization to novel objects, distractors, and environmental variations.

**Compliance With Llm Reviewing Policy:**

Affirmed.

**Final Justification:**

Thank the authors for their detailed response and explanation. I think this is a great work as well. In addition, I also look forward to the open-source release of the dataset and model, which will benefit the embodied AI community.

**Key Questions For Authors:**

1.  The codebook ablation (Table 14) shows that scaling from 256 to 512 categories yields no gains, yet precision deficiency remains the primary failure mode on dexterous tasks. Since the action decoder ultimately reconstructs continuous trajectories via flow matching, have the authors considered continuous latent alternatives (e.g., a VAE without quantization) that could preserve cross-modal alignment while avoiding the information loss inherent to discretization? Would increasing the discrete catagories more help learn the delicate controls?

**Limitations:**

yes

**Strengths And Weaknesses:**

Strengths

1. The paper is well-written and easy to follow. The three-stage pipeline is introduced in a logical flow, and the motivation for moving beyond unimodal latent representations is clearly articulated with appropriate positioning against prior work.

2. The evaluation spans six heterogeneous robot embodiments, over 120 manipulation tasks, and more than 14,000 real-world rollouts.

3. The framework's ability to leverage large-scale heterogeneous sources is a practical strength, and these data pipelines could serve as useful references for future cross-embodiment research. Additionally, demonstrating compatibility with both a large-scale backbone and a lightweight backbone.

Weakness.

1. UVMC encodes visual dynamics and robotic motion into a discrete codebook of 256 categories with 13 codes, yet the downstream action decoder reconstructs continuous action trajectories via flow matching. This introduces a representational bottleneck in which continuous motor commands must first be quantized, thereby inherently discarding sub-codebook-level information. The paper's own failure analysis supports this concern: precision deficiency is the primary failure mode, with success rates dropping to 15–25% on dexterous tasks. In addition, the codebook ablation shows that increasing the codebook size from 256 to 512 yields no further gains, suggesting that the discrete bottleneck may impose a ceiling on fine-grained control that cannot be resolved by simply scaling the codebook.

2.  The three-stage training pipeline requires approximately 80,000 GPU hours on 80 NVIDIA A100s for stage-1 and another 38,400 hours for stage-2, plus ~576 hours per embodiment for stage-3 fine-tuning. This cost is well beyond the reach of most academic research labs. Thus, it is difficult to assess whether the performance gains justify the additional compute overhead introduced by the UVMC pretraining stages.

---

> ### Author Rebuttal · Authors · 2026-03-30
>
> We thank `Reviewer CvQt` for the thoughtful feedback and for recognizing the clear motivation, strong experimental coverage, and practical value of XR-1 for cross-embodiment learning. Below, we provide a comprehensive response to the questions and concerns raised in the review.
>
> **1. On the Representational Bottleneck vs. Execution Precision.**
>
> >***1) Semantic Regularization for Cross-Embodiment:*** The primary goal of UVMC is to unify 6 heterogeneous robots into a shared latent space. A "lossless" continuous representation would inevitably capture embodiment-specific hardware noise (e.g., kinematic singularities, joint jitters), which hinders cross-platform alignment. By using a discrete bottleneck, we force the model to discard low-level geometric noise and focus on high-level action semantics (e.g., the intent to "pick" or "place"), which are universal across different morphologies.
> >***2)Hierarchical Decoupling of Semantics and Execution Precision:*** Our framework explicitly orthogonalizes high-level semantic planning (UVMC) and fine-grained geometric execution (Action Expert). While UVMC extracts robust, embodiment-agnostic motion primitives, the continuous Action Expert is responsible for local trajectory refinement. Consequently, precision limitations in dexterous tasks are primarily a function of visual perceptual resolution or the stochastic modeling capacity of the action head, rather than an artifact of the discrete bottleneck.  For more dexterous manipulation that requires higher action precision, we believe that fine-grained continuous motion generation should be primarily handled by the action head, potentially together with action post-processing strategies [1][2][3].
> >***3) Interpreting the Codebook Scaling (256 vs. 512):*** The fact that increasing the codebook size yields no further gains does not imply a "ceiling" of the discrete bottleneck. Instead, it suggests that 256 categories have already reached the intrinsic dimensionality required to cover the universal "Motion Alphabet" targeted by XR-1. Beyond this point, adding more codes merely introduces redundancy without adding semantic value.  We further analyze the semantic properties of UVMC in Appendix F through visualization of the Unified Vision-Motion Codes.
>
> **2. On Continuous Latent Alternatives (e.g., non-quantized VAEs).**
> We appreciate the reviewer’s suggestion to consider continuous latent variables. However, we believe continuous VAEs face fundamental challenges within our cross-embodiment paradigm:
>
> >***1) Limitations of Continuous VAEs:*** Continuous latent spaces inherently prioritize lossless reconstruction, which often leads to semantic over-smoothing when modeling highly heterogeneous data. In our case (6 disparate robots), a continuous VAE tends to preserve embodiment-specific information—such as hardware jitters and unique kinematic singularities—making it extremely difficult to align diverse physical signals into a unified representation.
> >***2) VQ-VAE as a Semantic Filter:*** In contrast, we utilize discretization as an intentional semantic bottleneck. By forcing heterogeneous motions into a discrete codebook, the model is compelled to discard low-level geometric noise and instead extract a shared "Motion Alphabet." This "symbolic regularization" is the key to achieving cross-embodiment normalization, ensuring that the latent space captures universal task semantics rather than hardware-specific artifacts.
>
> **3.On the computational cost of XR-1.**
> We agree that the full three-stage training pipeline is computationally expensive. Our view, however, is that Stage 1 should be regarded as a ***foundation-style pretraining*** step, similar to large-scale pretraining in LLMs or VLAs: although the upfront cost is high, once trained, it provides a reusable UVMC representation that can be directly adopted by the research community. To further reduce the practical barrier, we plan to ***open-source our code and pretrained models***, so that adapting XR-1 to a new robot would require only a small amount of Stage 3 fine-tuning. Moreover, the empirical gains from UVMC pretraining are substantial. In **Table 4**, the ablation on whether Stage 1 UVMC is included (Exp. 2/5) shows that performance improves from 28.3 to 66.7 when UVMC is used. We also analyze the scaling behavior of Stage 1 in Table 4 (Exp. 6-9), where performance improves from 29.2 to 65.0 as Stage 1 scaling increases. These results suggest that the additional pretraining cost brings significant and consistent benefits, rather than marginal improvements.
>
> [1] Mixture of Horizons in Action Chunking
> [2] Real-Time Execution of Action Chunking Flow Policies
> [3] VLASH: Real-Time VLAs via Future-State-Aware Asynchronous Inference
>
> We hope this resolves your concerns. Please feel free to let us know if you have any further questions, and we would be happy to address them.

---

> > ### Author Rebuttal · Reviewer_CvQt · 2026-04-03
> >
> > Thank the authors for their detailed response and explanation. I think this is a great work as well. In addition, I also look forward to the open-source release of the dataset and model, which will benefit the embodied AI community.

---

> > > ### Author Response · Authors · 2026-04-07
> > >
> > > Dear `Reviewer CvQt`,
> > > We sincerely thank you for your positive evaluation and support. We are pleased that our responses have addressed your concerns.
> > > We remain fully committed to the open-source release of our model and dataset to support the research community, and we will ensure all additional analyses are integrated into the final version of the manuscript. Thank you again for your valuable feedback.

---

### Official Review · Reviewer_D6Ze · 2026-03-13

**Soundness:** 3
**Presentation:** 3
**Significance:** 3
**Originality:** 3
**Overall Recommendation:** 5
**Confidence:** 4

**Summary:**

This paper proposes XR-1, a VLA framework centered on Unified Vision-Motion Codes (UVMC), a discrete latent representation jointly learned from visual dynamics and robotic motion via a dual-branch VQ-VAE. The key insight is that prior VLAs do not consider learning future vision, or treat vision and action in isolation, encoding either visual dynamics or action sequences separately. This fails to capture the multimodal correspondence between observation and motor execution. By learning a unified, embodiment-agnostic latent space and injecting it as intermediate supervision into a three-stage training paradigm, XR-1 achieves stronger performance and generalization across diverse embodiments and tasks.

**Compliance With Llm Reviewing Policy:**

Affirmed.

**Final Justification:**

Thank the authors for the detailed response. Overall, the paper is well-motivated and of high quality, with comprehensive and solid experiments. I hope the authors will consider updating the figures for better clarity, and incorporating the clarifications and additional analyses into the revised version. I believe this paper will bring insights to the direction of latent action learning. I encourage the authors to fully release the code, datasets, and model weights to benefit the robotics and embodied AI research community. I will raise my score to 5.

**Key Questions For Authors:**

- Motivation of architecture and training design choices. Why use future frames instead of other motion representations, e.g., key points [1, 2], trajectories [3], or optical flow [4]? These representations may be more directly aligned with visuomotor control. Why is a discrete VQ-VAE preferred over directly learning joint representations [5, 6, 7, 8]?
- Why is the data choice between Stage 1 and Stage 2 inconsistent? Why is only XR-D used for Stage 2 pretraining, while Stage 1 uses the full heterogeneous dataset? Will XR-D be open-sourced?
- Why separate Stage 2 and Stage 3? During deployment on a new embodiment after Stage 2, why not jointly optimize L_uvmc + L_act on task data (i.e., merge Stage 2 and Stage 3)? Could this yield additional performance and simplify the pipeline?
- Why not design UVMC to be explicitly predictive? UVMC encodes future information during Stage 1, but inference only relies on current observations. Why not design UVMC as an explicitly predictive representation? For example, by training the vision encoder to predict future frames from current observations rather than encoding future frames directly?
- Cross-embodiment and human-to-robot transfer. Embodiment-agnostic is one of the key motivations for joint vision-motion learning. How do visual dynamics learned from Ego4D actually benefit robotic manipulation? Could the authors provide additional evidence to validate the transfer?
- Confusing details in Figure 2. In Stage 2 of Figure 2, why are the learnable tokens $t$ not shown in the VLM input? What are learnable tokens? What do the pink and green tokens represent? There is no legend.
- Could the authors demonstrate that finetuned XR-1 is able to decode high-quality future frames from its predicted UVMC codes? This would validate the effectiveness of joint representation.
- Stage 3 finetuning takes 576 GPU hours on 8 A100 GPUs, which appears to be more time than the baselines. How does this compare to the finetuning cost of baselines on the same tasks?


[1] Li, Yi, et al. "Hamster: Hierarchical action models for open-world robot manipulation." *arXiv preprint arXiv:2502.05485* (2025).

[2] Wen, Chuan, et al. "Any-point trajectory modeling for policy learning." *arXiv preprint arXiv:2401.00025* (2023).

[3] Zheng, Ruijie, et al. "Tracevla: Visual trace prompting enhances spatial-temporal awareness for generalist robotic policies." arXiv preprint arXiv:2412.10345 (2024).

[4] Zhong, Zhide, et al. "Flowvla: Visual chain of thought-based motion reasoning for vision-language-action models." *arXiv preprint arXiv:2508.18269* (2025).

[5] Cen, Jun, et al. "Worldvla: Towards autoregressive action world model." *arXiv preprint arXiv:2506.21539* (2025).

[6] Li, Shuang, et al. "Unified video action model." arXiv preprint arXiv:2503.00200 (2025).

[7] Fang, Yu, et al. "Robotic VLA Benefits from Joint Learning with Motion Image Diffusion." arXiv preprint arXiv:2512.18007 (2025).

[8] Ye, Seonghyeon, et al. "World action models are zero-shot policies." arXiv preprint arXiv:2602.15922 (2026).

**Limitations:**

see above

**Strengths And Weaknesses:**

Strengths.
- Large-scale real-world evaluation. The paper conducts real-world rollouts across 6 embodiments and 120 manipulation tasks, including bimanual collaboration, dexterous manipulation, and long-horizon tasks.
- Clear motivation for joint vision-motion learning. The motivation is well explained and technically sound.
- Model-agnostic design. XR-1 is validated on two different backbones: pi0 and SwitchVLA, showing that UVMC can be applied across architectures.


Weaknesses.
- Concerns of unfair baseline comparisons. In the experiments, baselines are only finetuned on task data, while XR-1 additionally benefits from large-scale pretraining on XR-D. This conflates pretraining data advantage with architectural advantage, making it hard to attribute gains solely to UVMC. An additional experiment where XR-1 and other baselines (e.g., pi0, GR00T-N1.5) are trained on identical datasets is necessary to isolate the contribution of the proposed method.
- Unclear motivation for architectural design choices. Integrating motion dynamics into VLAs have multiple design choices in terms of architectures and motion representations, but the paper does not discuss existing designs and justify its specific decisions.
- High computational cost. Pretraining requires 76800 total A100 GPU hours (Stage 1 + Stage 2 combined) and finetuning per embodiment takes another 576 GPU hours. These costs are reported but not compared against baselines.
- Missing validation of future frame prediction. The finetuned XR-1 predicts unified vision-motion codes, which should be able to decode both visions and motions. However, there are no qualitative results for vision prediction. It is unclear whether the vision-motion codes predicted by finetuned XR-1 still preserves joint latent representation.
- No simulation benchmark evaluation. All experiments are real-world only, which limits reproducibility and makes comparison with a broader set of concurrent VLAs difficult.

---

> ### Author Rebuttal · Authors · 2026-03-30
>
> We thank `Reviewer D6Ze` for the positive assessment of XR-1. Below, we address the questions and concerns raised.
>
> **1. Concerns on Baseline Comparison**:
> > ***1) No Pretraining Data Advantage.*** The concern about a possible “pretraining data advantage” is not supported by the actual data scale. As shown in **Table R2_1**, XR-1 is pretrained on only a small fraction of the data used by recent baselines, yet still consistently outperforms them. This suggests that the gains come from the efficiency of UVMC rather than from larger-scale pretraining.
> > ***2) Unseen and similar embodiments.*** We further show that XR-1’s gains are architectural rather than due to data scale in two representative settings. On **Tien Kung 2.0** (**Table 2**), the embodiment is entirely unseen during pretraining, yet XR-1 reaches **72.0%** success, nearly doubling the best baseline (**40.8%**) under the same downstream data. On **Dual-Arm UR-5e** (**Figure 4**), the baselines use more in-domain pretraining data than XR-1, but XR-1 still outperforms `pi0.5` (**72% vs. 62%**).
> > ***3) UVMC Still Helps on Identical Training Data.*** To ensure a fair comparison, we compare **Table 4, Exp. 3 (`pi0`) and Exp. 5**, where both models are trained from scratch on the exact same task data. Under this identical-data setup, XR-1 with UVMC still achieves a **1.3x** improvement over the base `pi0` architecture, showing that the gain comes from the UVMC design itself.
>
> Table R2_1. Pretraining data scale and model size comparison.
> |Model| Data|Size|
> |---|---:|---:|
> | pi0 |10000h+OXE|3.5B|
> | GR00T-N1.5 |7000h+OXE|3B|
> | XR-1-S1 |1100h+OXE|1B|
> | XR-1-S2 |639h|4B|
>
> **2. Why UVMC, future frames, and VQ-VAE?**
> >***1)*** Existing VLAs often rely on either motion-only or vision-only representations. **UVMC** bridges this gap by aligning vision and motion latents in a shared space, making them more compatible with action decoding; see **Table 14** (Exp. 7/8/10).
> >***2)*** We use **future frames** through self-supervision because this scales naturally to large unlabeled datasets without the cost or noise of keypoints, trajectories, or optical flow. More importantly, XR-1 is a modular alignment framework: stronger vision models or learning objectives can be incorporated without changing the core paradigm.
> >***3)*** We use **VQ-VAE** because discretization enables both cross-embodiment normalization and semantic alignment. A shared discrete codebook clusters heterogeneous signals into a unified motion representation, filtering embodiment-specific noise and improving feature sharing across robots.
>
> **3. Computational Cost.** XR-1 is more compute-efficient than recent SOTA baselines during pretraining: as shown in **Table R2_1**, `pi0` and `GR00T-N1.5` use 3x-4x more pretraining data at comparable scale. For Stage 3, the 576 GPU-hours per embodiment is the same full-parameter fine-tuning cost used for all baselines in our setting, with the same number of training epochs.
>
> **4. Validation of Future Frame Prediction.** See the bottom of our project website: [https://xr-1-vla.github.io/](https://xr-1-vla.github.io/).
>
> **5. Simulation Experiments (CALVIN D_D).** Due to space limitations, we kindly refer the reviewer to Point 3 in our response to `Reviewer WMC9`.
>
> **6. Dataset Selection and Open-Sourcing.** **Stage 1 (1B)** uses the full heterogeneous dataset for broad priors, while **Stage 2 (4B)** uses the high-quality `XR-D` to balance performance and the higher compute cost of the larger model. `XR-D` will be fully open-sourced.
>
> **7. Why separate Stage 2 and Stage 3?** Stage 2 is pretrain for semantic alignment, while Stage 3 is for downstream adaptation. Once alignment is learned, $L_{uvmc}$ is no longer needed: jointly optimizing $L_{uvmc}+L_{act}$ gives similar performance to using $L_{act}$ alone, but with higher cost. We therefore use **Stage 3** as a lightweight adaptation stage trained only with $L_{act}$.
>
> **8. Why not design UVMC to be explicitly predictive?** Our goal is latent alignment between vision and action, rather than direct future prediction. We therefore use an encoding-based **Stage 1**: future prediction is inherently multimodal and can produce ambiguous representations, while joint encoding of current and future observations yields more stable motion-aligned UVMC tokens for **Stage 2** supervision.
>
> **9. Ego4D Benefits.** As shown in **Table 15**, removing `Ego4D` lowers the average success rate from **38.3** to **32.5**, confirming its benefit. More broadly, this shows that XR-1 can effectively leverage human video data, making it more scalable than VLAs trained only on action-labeled robot data.
>
> **10. Figure 2 Clarification.** We apologize for the confusion. In Stage 2, the pink tokens are learnable VLM tokens, and the green tokens are Stage-1 UVMC used as supervision. We will revise the legend to make this clearer.
>
> To respect the space limit, we focus on the key points and are happy to clarify any remaining questions.

---

> > ### Author Rebuttal · Reviewer_D6Ze · 2026-04-04
> >
> > Thank the authors for the response and clarifications.
> >
> > - Computational cost. Thanks for providing Table R2_1. Regarding Stage 3, do all baseline VLAs require the same number of training steps to converge? They may converge in fewer steps, in which case the fine-tuning cost of XR-1 would be higher in practice.
> > - Design choices. Regarding the selection of VQ-VAE, I would appreciate a more direct comparison between discrete and continuous latent spaces, preferably supported by results.
> >
> > This paper is overall well-motivated and of high quality, and I hope the clarifications above will be incorporated into the revised version. I also encourage the authors to consider updating the design and layout of the figures (e.g., Figure 2, Figure 4).

---

> > > ### Author Response · Authors · 2026-04-07
> > >
> > > **1. Convergence and Computational Cost.**
> > > > We thank the reviewer for the insightful concern on convergence fairness. We agree that different VLAs may exhibit different learning dynamics, and therefore do not assume identical convergence behavior across methods. To examine this directly, we evaluate `pi0.5` under extended fine-tuning budgets of **8, 12, and 16 epochs** on 10 representative Dual-Arm Franka tasks (Table RR_1). Results show that `pi0.5` improves from **30.5%** to **45.0%** and then **49.5%**, indicating clear diminishing returns as training increases. At the task level, simpler tasks become saturated early, while more dexterous tasks still benefit from additional training but remain challenging. Importantly, even at **16 epochs**, `pi0.5` still remains far below **XR-1 (75.0%)**, suggesting that the gap cannot be explained solely by insufficient training. We will clarify in the revision that our claim is not that all baselines converge equally fast, but that under a sufficiently extended Stage-3 budget, the strongest baseline already approaches its practical limit while XR-1 still retains a large advantage.
> > >
> > >
> > > Table RR_1: Dual-Arm Franka (10 Tasks) Performance
> > > | Task | pi0.5 (epoch=8) | pi0.5 (epoch=12) | pi0.5 (epoch=16) | XR-1 (epoch=16) |
> > > |:--|--:|--:|--:|--:|
> > > | DFR-MoveCupMilk   | 0 | 10 | 15 | 80 |
> > > | DFR-StackBowls    | 85 | 80 | 85 | 85 |
> > > | DFR-SweepTrash    | 30 | 50 | 60 | 75 |
> > > | DFR-TransferCup   | 5 | 30 | 40 | 90 |
> > > | DFR-MoveChopstick | 0 | 0 | 0 | 55 |
> > > | DFR-StackCubes    | 25 | 50 | 55 | 60 |
> > > | DFR-StackPlates   | 85 | 90 | 90 | 90 |
> > > | DFR-CleanTable    | 60 | 85 | 85 | 90 |
> > > | DFR-HangCupHolder | 15 | 40 | 45 | 65 |
> > > | DFR-HangTowelRack | 0 | 15 | 20 | 60 |
> > > | **Avg Success Rate** | 30.5% | 45.0% | 49.5%  | 75.0% |
> > >
> > >
> > >
> > > **2. Direct Comparison: Discrete (VQ-VAE) vs. Continuous (VAE) Latent Spaces.**
> > > We thank the reviewer for this insightful question. To provide a more direct comparison between discrete and continuous latent spaces, we conduct a strictly controlled ablation study in which all stage-1 components and training settings are kept unchanged, and only the latent model is replaced, i.e., **VQ-VAE** versus **VAE**. On the 6 Dual-Arm UR tasks, we compare the two designs from three aspects: **(1) stage-1 training dynamics, (2) qualitative visualization of future-frame prediction, and (3) downstream task success rates.** Overall, these results consistently support our choice of VQ-VAE over VAE.
> > >
> > > The corresponding qualitative results have also been added to the bottom of our project website: [https://xr-1-vla.github.io/](https://xr-1-vla.github.io/).
> > >
> > > > ***1) Stage-1 Training Dynamics:*** We compare both the **action reconstruction loss** and the **vision reconstruction loss**. VQ-VAE consistently outperforms VAE on both metrics. Specifically, the final action reconstruction loss is **0.0823 vs. 0.1277**, and the vision reconstruction loss is **0.0356 vs. 0.0549**. These results show that, under the same training setup, VQ-VAE learns a more effective latent representation and achieves better convergence. This finding further suggests that, compared with a continuous latent space, discrete latent tokens provide a more structured and decoder-friendly representation.
> > > > ***2) Stage-1 Visualization of Future-Frame Prediction:*** Beyond the quantitative results, we further provide qualitative visualizations of future-frame prediction. VQ-VAE produces reconstructions that are consistently sharper and preserve key objects and scene dynamics more faithfully than VAE. For example, in the task `DUR-SweepTrash`, the VAE reconstruction fails to preserve the broom clearly, whereas VQ-VAE still retains its main structural features. This qualitative evidence is consistent with the lower reconstruction losses and further supports the advantage of VQ-VAE in preserving visually and dynamically relevant information.
> > > > ***3) Evaluation on 6 Dual-Arm UR Tasks:*** We further evaluate the two latent designs on downstream manipulation performance. Replacing VAE with VQ-VAE leads to consistently higher success rates across all 6 tasks, improving the average success rate from **52.5%** to **66.7%**. This result (Table RR_2) shows that the stronger representation learned in stage 1 translates into clear gains in downstream sequential manipulation.
> > >
> > > Table RR_2: Dual-Arm UR (6 Tasks) Performance
> > >
> > > | Method | DUR-CleanTable | DUR-FindTapeBasket | DUR-MoveCupMilk | DUR-StackBowls | DUR-SweepTrash | DUR-TransCupHolder | Avg. |
> > > |---|---:|---:|---:|---:|---:|---:|---:|
> > > |baseline|0|50|20|55|0|45|28.3|
> > > |baseline+VAE|40|70|45|65|45|50|52.5|
> > > |baseline+VQ-VAE|50|75|65|80|60|70| 66.7 |
> > >
> > > Dear `Reviewer D6Ze`,
> > > Overall, we believe our additional convergence analyses and VQ-VAE ablation experiments fully address your concerns regarding fairness and design choices, and we would sincerely appreciate a ***reconsideration of the score*** based on these results.

---

### Official Review · Reviewer_WMC9 · 2026-03-18

**Soundness:** 3
**Presentation:** 2
**Significance:** 3
**Originality:** 3
**Overall Recommendation:** 5
**Confidence:** 5

**Summary:**

Existing VLAs face a domain gap when processing heterogeneous data sources. Therefore, this paper proposes XR-1, which mainly focuses on the research of cross-embodiment representation sharing and representation alignment in image-to-action tasks. Its core is the Unified Vision-Motion Codes (UVMC). Through a dual-branch VQVAE, UVMC maps vision and actions into a unified discrete latent space, serving as the initial stage of VLA training. The authors conducted extensive real-robot experiments on 6 robot configurations, demonstrating the reliability and cross-embodiment capability of XR-1.

**Compliance With Llm Reviewing Policy:**

Affirmed.

**Final Justification:**

Thank the authors for their detailed response. I still consider this an excellent piece of work. I hope the authors can incorporate these additional experiments into the revised version, and I will raise my score to 5.

By the way, I am looking forward to the open-source release of the dataset and model, which will be highly beneficial to the embodied AI community.

**Key Questions For Authors:**

See Weakness

**Limitations:**

I couldn't find the limitations section. In my view, the limitations are largely common drawbacks of the VLA paradigm. That said, it would be significantly better if the training dynamics in the first stage could be clearly analyzed.

**Strengths And Weaknesses:**

## Strengths
1. Compared with previous methods that encode visual features and action sequences separately, UVMC establishes a causal connection between visual changes and physical actions through joint encoding, which is an innovative VLA training approach.

2. Due to the design of cross-ontology general representations, XR-1 can directly leverage unlabeled human videos (e.g., Ego4D) to learn dynamic priors of the physical world, which is an excellent solution to the problem of data scarcity.

3. Extensive and detailed real-robot experiments were conducted, covering multiple scenarios, multiple ontologies, and multiple experimental settings (zero-shot/few-shot), which can fully prove the effectiveness of the XR-1 design.

4. The experimental results of the Light version are also meaningful. Based on Florence-2, a small-scale VLM, it can prove the effectiveness of this technical route under different computing power scales.

## Weaknesses and Questions

1. The design details of the first stage lack excellent ablation studies or analyses. However, considering the high pre-training cost, I think this is understandable, but I still have some questions:


  - Why do vision and actions share the same codebook? What differences would there be if they are trained separately and then aligned?

  - Is the cross-modal alignment of vision and actions completely achieved through the KL loss term in Equation 3?

  - Is there a risk of codebook collapse during the first-stage training of XR-1? Is this stage of training stable, or does it require multiple hyperparameter tuning?

  - All loss weights in line 257 are set to 1; is it necessary to adjust the weight of the KL term?

  - There is a lack of more ablation studies or visualization analyses on the internal design details of Stage 1. Currently, all analyses are from the perspective of success rate. Can similar results be achieved by using the same training method to separately train representations for observations and actions?

2. I highly recognize the extensive real-robot experiments provided by the authors, but I believe that a small number of simulation experiments to verify the effectiveness of the method and conduct unified benchmark comparisons can further improve the richness and reliability of the paper.

3. Do the authors plan to open-source the dataset they collected?

4. Lines 367-371:
        e.g., in TK2-MoveCupSauce, it reaches 70% versus 60% for π0. These results indicate that UVMC effectively encodes embodiment-agnostic dynamics into a shared latent space, enabling efficient transfer of prior knowledge to novel robotic platforms.
        Why can this demonstrate cross-ontology transfer performance? Intuitively, this only shows an improvement in success rate.


In summary, this paper involves enormous workload, with rigorous methodology and impressive experimental results. It takes an important step forward in how to use heterogeneous data to improve the accuracy of robot actions.

---

> ### Author Rebuttal · Authors · 2026-03-30
>
> We thank `Reviewer WMC9` for the constructive feedback and for recognizing the novelty of UVMC, the practical value of learning from heterogeneous data sources, and the strong experimental results of XR-1 across multiple robot embodiments and settings. Below, we provide a comprehensive response to the weaknesses and questions raised.
>
> **1. More ablation study and visulazation of analysis on UVMC.** We ablated both ***shared and separate codebook*** designs and found that both are viable. The corresponding results are reported in **Table 14** (Exp. 9/10), with discussion in Lines 1188-1192. We also studied ***vision-only and motion-only codebooks***; these results are shown in Table 14 (Exp. 7-8/10) and discussed in Lines 1184-1186. In addition, Appendix F (Lines 1309-1521) provides ***visualization-based analysis*** of UVMC using nearest-neighbor retrieval and t-SNE.
>
> **2. KL loss for modality alignment.**
> > ***1) Alignment Mechanism:*** Cross-modal alignment is achieved through the joint reconstruction of vision and motion. The KL term in Eq. 3 acts as a regularizer to prevent the latent space from becoming arbitrarily high-variance, ensuring a smooth transition between vision and motion priors.
> > ***2) Stability and KL Sensitivity:*** Our experiments show that $W_{KL}=1$ provides the optimal balance. We observed that while a very high penalty (e.g., $W_{KL}=10$) can lead to codebook under-utilization on limited datasets—where the model might prioritize satisfying the $KL$ constraint over encoding complex details—setting $W_{KL}=1$ ensures high codebook activity and robust reconstruction across all data scales.
> > ***3) Large-Scale Pretraining Robustness:*** Crucially, during our actual large-scale pretraining (Stage-1), the immense diversity of the heterogeneous data (Ego4D, OXE, Robomind, XR-D) naturally prevents "hard-coding" shortcuts and codebook collapse. We found that the training is remarkably stable under our current configuration, requiring no specialized hyperparameter scheduling to maintain codebook vitality.
>
> **3. Simulation Experiments (CALVIN $D \to D$).** To address the reviewer’s concern on simulation validation, we include results on the challenging CALVIN $D \to D$ benchmark for instruction-conditioned long-horizon manipulation. This setting is particularly sensitive to error accumulation and resulting distribution drift. As shown in **Table R1_1**, XR-1 achieves a new state-of-the-art success rate of **4.256**, outperforming strong baselines such as `pi0.5` (**3.885**) and `qwengr00t` (**3.786**). Notably, XR-1 still attains **0.741** success at the 5th sub-task, showing stronger long-horizon robustness and better resistance to distribution drift.
>
>     Table R1_1. Comparison of XR-1 with baselines on CALVIN D→D.
>     | CALVIN D→D                | 1     | 2     | 3     | 4     | 5      | Success rate |
>     |---------------------------|-------|-------|-------|-------|--------|--------|
>     | OpenVLA                   | 0.716 | 0.385 | 0.180 | 0.088 | 0.0412 | 1.411  |
>     | RDT-1B                    | 0.757 | 0.495 | 0.359 | 0.243 | 0.184  | 2.038  |
>     | pi0                       | 0.848 | 0.704 | 0.559 | 0.466 | 0.377  | 2.954  |
>     | qwenpi0                   | 0.909 | 0.795 | 0.696 | 0.622 | 0.554  | 3.576  |
>     | qwengr00t                 | 0.925 | 0.839 | 0.744 | 0.679 | 0.599  | 3.786  |
>     | pi0.5                     | 0.925 | 0.840 | 0.766 | 0.710 | 0.644  | 3.885  |
>     | XR-1(ours)                | 0.964 | 0.908 | 0.845 | 0.798 | 0.741  | 4.256  |
>
> **4. Open-Sourcing the XR-D Dataset.** We will release the XR-D dataset and plan to provide an even larger version than the one described in the paper, in order to further support academic research.
>
>
> **5. Cross-Embodiment Gains (TK2-MoveCupSauce).**
> > ***1) Direct Comparative Baseline:*** Since XR-1 adds UVMC to a backbone consistent with $\pi_0$, the gap to $\pi_0$ directly reflects the benefit of UVMC. On TK2-MoveCupSauce, XR-1 improves success rate from **60%** to **70%**.
> >***2) Semantic Skill Transfer:*** We attribute this gain to cross-modal semantic alignment. Although **TK2 data is absent from XR-D pretraining**, UVMC captures transferable skill semantics such as pick-and-place from heterogeneous robots and human videos.
> >***3) Evidence of Alignment:*** As shown in **Appendix F, Figure 12**, UVMC clusters functionally similar actions into a shared latent space across embodiments, enabling more effective finetuning on new hardware such as Tien Kung 2.0.
>
> **6. Training Dynamics of UVMC and Limitations.** We have substantially expanded the analysis of UVMC training dynamics and limitations, and provide additional results on our project website: [https://xr-1-vla.github.io/](https://xr-1-vla.github.io/).
>
> To respect the space limit, we focus on the key points and are happy to clarify any remaining questions.

---

> > ### Author Rebuttal · Reviewer_WMC9 · 2026-04-02
> >
> > Thank the authors for their detailed response. I still consider this an excellent piece of work. I hope the authors can incorporate these additional experiments into the revised version, and I will raise my score to 5.
> >
> > By the way, I am looking forward to the open-source release of the dataset and model, which will be highly beneficial to the embodied AI community.

---

> > > ### Author Response · Authors · 2026-04-07
> > >
> > > Dear `Reviewer WMC9`,
> > > We sincerely thank you for your positive evaluation and for increasing the score. We will ensure that all additional experiments and analyses are integrated into the final camera-ready version. We also remain fully committed to open-sourcing our model and dataset to support the embodied AI community. We appreciate your valuable feedback, which has significantly improved the quality of our work.

---

### Decision · Program_Chairs · 2026-04-30

**Decision:**

Accept (spotlight)

**Comment:**

All reviewers agree that this is high-quality work, which is of great interest to embodied AI community.
Especially the extensive empirical evaluation on real-world robots and the authors to open source their dataset and model stand out.
I recommend accepting this paper without any reservations.